# Stain-Free Sperm Analysis and Selection for Intracytoplasmic Sperm Injection Complying with WHO Strict Normal Criteria

**DOI:** 10.3390/biomedicines11102614

**Published:** 2023-09-23

**Authors:** Yulia Michailov, Luba Nemerovsky, Yehudith Ghetler, Maya Finkelstein, Oshrat Schonberger, Amir Wiser, Arie Raziel, Bozhena Saar-Ryss, Ido Ben-Ami, Olga Kaplanski, Netanella Miller, Einat Haikin Herzberger, Yardena Mashiach Friedler, Tali Levitas-Djerbi, Eden Amsalem, Natalia Umanski, Valeria Tamadaev, Yaniv S. Ovadia, Aharon Peretz, Gilat Sacks, Nava Dekel, Odelya Zaken, Mattan Levi

**Affiliations:** 1Obstetrics and Gynecology Department, Barzilai University Medical Center, Ashkelon 7830604, Israel; bozhena.ryss@gmail.com (B.S.-R.); eden88amsalem@gmail.com (E.A.); natasha.umansky@gmail.com (N.U.); yaniv.ovadia@mail.huji.ac.il (Y.S.O.); 2Faculty of Health Sciences, Ben Gurion University of the Negev, Beer Sheva 8410501, Israel; 3Faculty of Medicine, Tel Aviv University, Tel Aviv 6997801, Israel; lnemerovsky@gmail.com (L.N.); mayaf@wmc.gov.il (M.F.); amir.wiser@gmail.com (A.W.); arier@wmc.gov.il (A.R.); olga.kaplanski@gmail.com (O.K.); millerne@me.com (N.M.); einat.haikin@gmail.com (E.H.H.); dryardena@gmail.com (Y.M.F.); tali.levitas@gmail.com (T.L.-D.); mattanlevi@gmail.com (M.L.); 4IVF Unit, Department of Obstetrics and Gynecology, Meir Medical Center, Kfar Saba 4428163, Israel; yghetler@gmail.com; 5IVF Unit, Wolfson Medical Center, Holon 5822012, Israel; 6IVF Unit, Shaare Zedek Medical Center, Jerusalem 9103102, Israel; oshrativf@szmc.org.il (O.S.); idoba@szmc.org.il (I.B.-A.); aharonperetz@gmail.com (A.P.); kupgil@gmail.com (G.S.); dekel.nava@gmail.com (N.D.); odelyazaken@gmail.com (O.Z.); 7Faculty of Medicine, Hebrew University, Jerusalem 9112102, Israel

**Keywords:** sperm selection, IVF ICSI, quantitative phase microscopy (QPM), WHO2021

## Abstract

This multi-center study evaluated a novel microscope system capable of quantitative phase microscopy (QPM) for label-free sperm-cell selection for intracytoplasmic sperm injection (ICSI). Seventy-three patients were enrolled in four in vitro fertilization (IVF) units, where senior embryologists were asked to select 11 apparently normal and 11 overtly abnormal sperm cells, in accordance with current clinical practice, using a micromanipulator and 60× bright field microscopy. Following sperm selection and imaging via QPM, the individual sperm cell was chemically stained per World Health Organization (WHO) 2021 protocols and imaged via bright field microscopy for subsequent manual measurements by embryologists who were blinded to the QPM measurements. A comparison of the two modalities resulted in mean differences of 0.18 µm (CI −0.442–0.808 µm, 95%, STD—0.32 µm) for head length, −0.26 µm (CI −0.86–0.33 µm, 95%, STD—0.29 µm) for head width, 0.17 (CI −0.12–0.478, 95%, STD—0.15) for length–width ratio and 5.7 for acrosome–head area ratio (CI −12.81–24.33, 95%, STD—9.6). The repeatability of the measurements was significantly higher in the QPM modality. Surprisingly, only 19% of the subjectively pre-selected normal cells were found to be normal according to the WHO2021 criteria. The measurements of cells imaged stain-free through QPM were found to be in good agreement with the measurements performed on the reference method of stained cells imaged through bright field microscopy. QPM is non-toxic and non-invasive and can improve the clinical effectiveness of ICSI by choosing sperm cells that meet the strict criteria of the WHO2021.

## 1. Introduction

Infertility, defined as the inability to achieve pregnancy after 12 months of regular and unprotected sexual intercourse, is an international health concern which affects ~12% of couples worldwide. Infertility comes with profound physical, emotional and societal implications for both parties. While traditional infertility treatments and investigations have focused on female factors, recent research has highlighted the crucial role of male infertility in this complex issue and has shown that over the past few decades, there has been a discernible increase in reported cases of male infertility worldwide. Factors such as lifestyle choices, environmental exposures, alcohol and drug use, and an in-creased tendency to wait until later in life to begin having children can all contribute to infertility. A growing awareness of the issue has driven the need for a comprehensive understanding of its causes and for finding solutions [1].

Intracytoplasmic sperm injection (ICSI) is a specialized assisted reproductive technology (ART) procedure used to assist in cases of infertility for which male factors are the primary cause [2]. In ICSI, a single sperm is selected and injected into an oocyte to facilitate fertilization. In cases of male-factor infertility, ICSI significantly improved the chances of successful fertilization and pregnancy [3]. However, sperm selection during ICSI is primarily based on manual and subjective morphological assessment and needs to be standardized for uniformity and concomitance between embryologists [4]. Sperm quality, including both motility and morphology, is a crucial factor in achieving fertilization and successful pregnancy [5]. Studies have shown that a high rate of sperm motility and normal sperm morphology in a given sample positively correlate with fertilization and pregnancy rates during ICSI [6,7]. Sperm selection during ICSI typically involves an evaluation of sperm shape, size and structure as well as motility parameters under low (X20) magnification. Sperm analysis involves the fixation and staining of cells before observation under a higher magnification (X100). In this analysis, the criteria for normal morphology typically include parameters such as head size and shape, acrosome integrity, the presence of head vacuoles, and neck and tail length and curvature based on WHO2021 criteria. However, the method has significant limitations as it cannot detect sperm DNA damage, assess organelles, or perform quantitative calculations for movement trajectory. Perhaps most limiting are that cell staining is cytotoxic and that cells that are stained cannot be used in ICSI procedures.

The inability to assess certain aspects like sperm DNA damage and organelles under low magnification has led to new strategies, such as intracytoplasmic morphologically selected sperm injection (IMSI). IMSI is a method in which high magnification (×1000–10,000) is used to select sperm cells of the best morphological grade. This technique allows for a better assessment of sperm morphology and a better visualization of the vacuoles in the sperm head [8]. Initially, this method was reported to achieve better clinical pregnancy rates [9,10]. However, the later literature regarding this time-consuming and expensive approach is contradictory [11,12]. Even in cases where sperm exhibit normal morphology, the presence of DNA damage or other abnormalities can negatively impact fertilization and embryo development [13]. One of the significant challenges embryologists face is the limited ability to accurately assess full sperm morphology without using staining techniques, which render sperm unusable in ICSI. While there are commercial products such as PICSI dishes, SpermCatch and SpermSlow that offer some assistance in sperm selection by evaluating the expression of hyaluronic acid (HA) receptors [14], unfortunately, the data concerning the utilization of HA in ICSI are still controversial [15]. Innovative microfluidic techniques have emerged, enabling single-cell analysis of sperm for diagnostic purposes [16]. Another promising method, magnetic-activated cell sorting (MACS), which relies on membrane integrity, has demonstrated improved pregnancy rates compared to traditional density centrifugation and swim-up techniques [17]. However, some studies have failed to show any significant enhancement in clinical outcomes with MACS-selected sperm [18]. Emerging technologies, such as artificial intelligence (AI) and deep machine learning, have gained traction in recent years. These technologies can assess progressive motility parameters to aid embryologists in sperm selection without causing damage or exposing the sperm to chemical stains [14,19,20]. In recent years, our group has been developing a stain-free method for sperm classification based on AI, which will allow embryologists to evaluate a single sperm [21]. This method suggests applying virtual label-free imaging of biological samples using a deep learning approach, called HoloStain, on ICSI procedures. This non-intrusive method allows our system to convert unstained images to “stain-like” ones using a deep neural network [22,23] and allows for the measurement of sperm internal morphology without using chemical staining and real-time application of the WHO2021 criteria of normal sperm morphology [24].

Based on our previous findings and with the goal of incorporating QPM into the sperm assessment stage, we designed a clinical study that evaluated, a novel quantitative phase microscopy system in clinical IVF labs. The system incorporates the advanced technology of QPM and allows for label-free selection of sperm cells to be used in intracytoplasmic sperm injection (ICSI). The system records an integral refractive index map of the cells using interferometric imaging [25] by superimposing a monochromatic light beam interacting with the sample with a mutually coherent reference beam. The resulting profile is a topographic map, which indicates the dry mass surface density of the cell at all of its points. In sperm cells, this topographic map can facilitate the real-time application of the WHO2021 criteria for morphological sperm analysis. This analysis includes visualizing internal organelles as if the cells had been stained but without the use of cytotoxic stains. Through the use of optical computed tomography, a full 3D refractive index image can be generated by recording the interferometric images from multiple viewpoints [26]. To date, precise information regarding the fine organelle morphology of sperm cells is unavailable to embryologists during ICSI procedures. However, QPM allows the system to measure the cell refractive index distribution, using an intrinsic contrast mechanism of the cells to allow quick and reliable measurements of its internal morphology without using chemical staining, making it possible to apply the WHO2021 criteria of normal morphology in real time during ICSI procedures [24].

## 2. Materials and Methods

### 2.1. Study Design

This study is a double-blinded comparison between the automated measurements of the QPM technology used in the QPM system and the corresponding measurements of the same sperm cells after they were chemically stained in accordance with WHO human sperm processing protocols [25] and acquired via standard bright field microscopy (BFM) imaging [27]. Imaging and analysis were performed by senior embryologists as the human operators. The objective of this comparison is to explore the agreement between the QPM and a reference method.

Since the study compares morphological measurements of live, motile sperm cells, these parameters can deviate somewhat when derived from different intervals of the same QPM recording. Therefore, an additional objective of this study was to assess the repeatability of morphological measurements of the same sperm cells when taken from two different intervals of the same recording. A similar repeatability assessment was performed between two sets of BFM images of sperm cells and manually measured by the readers. A comparison between the repeatability of the QPM and the reference method was also performed.

Since the QPM system is designed to aid the embryologist in confirming compliance of specific sperm cells with consensus morphometric criteria, it is imperative to verify that the QPM system can reliably measure such parameters even with non-WHO2021-compliant sperm cells. For this reason, the sample group was a priori divided into an equal number of “normal morphology” and “abnormal morphology” sperm cells, as subjectively decided by the embryologists.

Another study endpoint was to assess the accuracy of the subjective classification of sperm cells, as performed by embryologists, in reference to WHO2021 morphometric criteria. Put simply, how likely would a sperm cell that an embryologist identifies as displaying “normal morphology” be verified as having such a morphology through QPM virtual staining and chemical staining.

The multi-center study was approved by the local institutional Helsinki committee of each participating center (Table 1). The study population included sperm samples provided by the clinic’s patients (recruited by the investigators from the clinic’s database). Healthy volunteering males, aged > 18, admitted to the IVF unit for infertility treatment with ICSI, were included in the study. All included subjects provided written informed consent after the research protocol was explained to them in detail, including the use of their de-identified samples. This study was a non-interventional study, no adverse events were anticipated and no safety analysis was defined.

Subjects diagnosed with severe oligozoospermia or documented presence of infectious disease transmitted via sperm fluids were excluded. A total of 75 participants who met the inclusion criteria were enrolled in 4 IVF units in Israel. No clinical decisions were made based on the QPM results. Imaged sperm cells were not used for injection, no oocytes were involved in this study and sperm samples were discarded after imaging.

After standard preparation of the sperm samples for ICSI treatment (see sperm processing section), 1 microliter of sperm was transferred to a de-identified tube and afterwards subjected to analysis by a single senior embryologist of each participating medical center. Twenty-two spermatozoa were individually selected and graded: eleven apparently normal (suitable for ICSI, hereinafter referred to as “a priori-normal”) and eleven overtly abnormal sperm cells (unsuitable for ICSI, hereinafter referred to as “a priori-abnormal”), in accordance with current clinical practice. Following the grading of each of the 22-sperm cell selection and imaging via QPM and BFM, each sperm cell was chemically stained with Quick Stain (QS) cat number 01-939-1U, Biological Industries Israel) per WHO reference protocol and imaged via bright field microscopy. Additionally, the BFM images were reviewed and evaluated for eligibility by expert embryologists. Cells that had at least one adequate BFM image for evaluation were included in a further analysis. Measurements of the chemically stained BFM images were calculated using the software marking tool (1.0) incorporated with QPM: (1) by built-in software automatic algorithm; (2) manually by QART embryologists blinded to the QPM automatic measurements. The measurements derived from both methods were compared to known data according to the WHO2021 manual for sperm size and shape (Table 2).

### 2.2. Sperm Processing

The seminal fluid of each subject was placed on top of a 40 and 80% silicon bead gradient (ORIGIO, Målov, Denmark) and centrifuged for 20 min at 500RCF. After centrifugation, the supernatant was discarded and the pellet with the sperm cells was washed with 5 mL of MHM-C (ref 90166, Irvine Scientific, Santa Ana, CA, USA) for 5 min at 500RCF. After the wash, the supernatant was discarded. An analysis glass culture dish (FD5040-100, World Precision Instruments, Sarasota, FL, USA) was prepared as follows: 8 drops of the 5 µL PVP7% (ART-4005-A, ORIGIO), 2 drops of the 1 µL sperm sample), 1 drop of the 5 µL QS mixture (20 µL QS with 80 µL PVP 7%) according to manufacturer instructions. The drops were covered with 3 mL of mineral oil (ref 10029, Vitrolife, Göteborg, Sweden).

### 2.3. Equipment

Each spermatozoon was collected from an MHM-C drop into microinjection pipette (30′Bend, LISR, Thebarton, Australia) installed on micromanipulator (MN-4, NARISHIGE, Tokyo, Japan) with matching controlling system and injector (MMO-4 and IM-11-2, respectively, NARISHIGE) and captured upon movement stabilization. Sperm images were acquired individually using a camera (Olympus U-TVO-35XC-2) installed on an Olympus CKX53 microscope with an X60/1.42oil lens (ꝏ/0.17/OFN26.5 Olympus, Tokyo, Japan).

### 2.4. QPM Technological Principles

The system for sperm stain-free evaluation was described in our group’s previous publications [22,23]. In this study, a commercial version of the interferometric microscope system, called Q300™, was provided by the study sponsor. The technological principle of the current study is identical to that published in previous studies, and the same QPM system was installed in each IVF unit, participating in the study. This clinical study evaluated, for the first time, the Q300™ in a clinical setting. The device incorporates the advanced technology of QPM and allows label-free sperm cell selection for ICSI. This technology enables the measurement of internal organelles and an objective automatic assessment of embryologist-selected sperm cells according to the WHO2021 guidelines in real time. The system does so without chemical staining by recording the integral refractive index of the cells using interferometric imaging. An image of the QPM system and its results screen is shown in Figure 1.

### 2.5. Statistical Analysis and Calculations

The study compared two modalities: the Investigational QPM System and the Reference Staining Methods. To explore the agreement between QPM and the reference method, Bland–Altman analysis figures were generated and 95% confidence intervals (1.96XSD) of the agreements were calculated. Estimated bias (defined as the mean difference between the 2 methods ± SD) was calculated. The maximum allowed limits were calculated and presented in the plots.

A repeatability evaluation was performed for the QPM system and the reference method with respect to all outcome measures reported by the device (head length, head width, length-to-width ratio and acrosomal-area-to-head-area ratio). For each modality, two measurements were taken per parameter per cell, the agreement between the two measurements for each modality was also presented using Bland–Altman plots and the 95% confidence intervals (1.96XSD) of agreements were calculated. Estimated bias (defined as the mean difference between the 2 methods ± SD) was calculated. To evaluate the difference in sperm cell classification, a sensitivity analysis between the two modalities was performed.

## 3. Results

In total, 1451 sperm cells were imaged across 73 subjects using the QPM device. Out of the 1451 sperm cells imaged by the device, 631 sperm cells were excluded by the QPM and an additional 820 sperm cells were excluded by the reference method following core lab evaluation. The reasons for sperm cell evaluation exclusion are detailed in Table 3. The baseline clinical characteristics and subject demographics are displayed in Table 4. The analysis comparing the Q300™ system results and the reference method results was performed with sperm cells for which measurements from both modalities were available, leaving a total of 326 sperm cells from 44 subjects. Of those 326 sperm cells, 199 sperm cells had an a priori-normal evaluation and 127 sperm cells had an a priori-abnormal evaluation. The average age of the subjects was 38.5 ± 7.4 years.

The agreement between the QPM and the reference method was evaluated for 326 sperm cells. The average characteristic values (the median over the mean measurements between the reference and QPM results, for each cell) of the sperm cells evaluated were 4.7 µm head length, 3.2 µm head width, 1.4 length–width ratio and 49% acrosome-to-head ratio. Agreement of the two modalities resulted in biases of 0.18 µm (CI −0.442–0.808 µm, 95%, STD—0.32 µm) for head length, −0.26 µm (CI −0.86–0.33 µm, 95%, STD—0.29 µm) for head width, 0.17 (CI −0.12–0.478, 95%, STD—0.15) for length–width ratio and 5.7 for acrosome–head area ratio (CI −12.81–24.33, 95%, STD—9.6). Bland–Altman plots are presented in Figure 2.

Maximum allowed limit calculation: the maximum allowed limits were calculated using the standard deviation of the reference method repeatability analysis. In each graph, the limit was calculated per the correspondent parameter (length, width, length to width and acrosome area to head area) of the reference method repeatability analysis.

The maximum allowed limit calculation:Diff=mean value ±1.96·agreement STD±AL
$$AL=\sqrt{{reference\;repeatability\;STD}^{2}+0.1\cdot {agreement\;STD}^{2}}$$

For example, in the length plot, the maximum allowed limits were located at the following:Diff=length difference mean value ±1.96·length different agreement STD±AL
$$AL=\sqrt{{length\;difference\;markers\;R\&R\;STD}^{2}+0.1\cdot length\;difference\;{agreement\;STD}^{2}}$$

The secondary endpoint, a comparison between the QPM repeatability and the reference method repeatability, was performed. For the QPM repeatability evaluation, 114 sperm cells were analyzed for their length, width, length to width ratio and acrosomal area to head area ratio. Repeatability of the reference method was determined on 824 sperm cells. The repeatability agreement of the QPM vs. the reference method for each parameter resulted in biases of −0.017 µm (CI −0.28–0.12 µm, 95%, STD—0.13 µm) vs. 0.32 µm (CI −0.3–0.95 µm, 95%, STD—0.32 µm) for head length, respectively; −0.012 µm (CI −0.22–0.09 µm, 95%, STD—0.1 µm) vs. 0.25 µm (CI −0.33–0.83 µm, 95%, STD—0.29 µm) for head width; 0.0006 (CI −0.102–0.05, 95%, STD—0.05) vs. −0.03 (CI −0.3–0.25 95%, STD—0.14) for length–width ratio; and −0.73 (CI −15.1–6.7, 95%, STD—7.49) vs. 0.65 (CI −17.5–18.8, 95%, STD—9.29) for acrosome–head area ratio. Bland–Altman plots are presented in Figure 3.

The classification agreement was determined on the same 326 cells used for the agreement evaluation. In total, 199 (61%) of the cells were a priori classified as normal and 127 (38.9%) of the cells were a priori classified as abnormal by the embryologists.

Of the 199 cells which were a priori evaluated as displaying normal morphology by manual selection,

(1)25% (50) were later classified by the reference method as compliant with WHO2021 guidelines;(2)19% (38) were later classified by the Q300™ as compliant with WHO2021 guidelines.

The sensitivity analysis, performed between the results generated by the Q300™ and reference methods, aiming to reflect the ability of the QPM device to detect the abnormal (non-complaint with WHO2021 guidelines) sperm cells, resulted in a value of 88.6%; the accuracy of the method was 73.4%. A sub-analysis of sensitivity and accuracy performed only on the population which were a priori evaluated as normal resulted in 85.2% sensitivity and 71.5% accuracy.

## 4. Discussion

This study demonstrated that QPM technology can be successfully used in an IVF laboratory environment. Agreement between the QPM and the reference method was adequate and as can be seen in the Bland–Altman plots; the confidence intervals were well within the maximum allowed limits for all evaluated parameters. Agreement between the QPM and the reference method was performed for the sperm head length, head width, head length–width ratio and acrosome-to-head ratio. A sensitivity analysis, performed between the results derived from the QPM and the reference method, aiming to reflect the ability of the QPM to detect the abnormal (non-complaint with WHO2021) sperm cells, resulted in a value of 88.6%; the accuracy of the method was 73.4%. A-sub analysis of sensitivity and accuracy performed only on the population which were a priori evaluated as normal by manual selection resulted in 85.2% sensitivity and 71.5% accuracy. Moreover, the repeatability of measurements was significantly higher in the QPM modality compared to that in the reference.

Another potentially important finding from the study was that less than 25% of the cells selected by embryologists through manual and subjective selection were later confirmed to be compliant with the WHO2021 criteria. This was evident both when comparing the manually selected sperm cells via the results derived by QPM and the reference method results. As shown in the sensitivity analysis performed between the two modalities for detecting abnormal sperm cells, the QPM results were satisfying, with a high degree of sensitivity and accuracy. This is arguably the most useful feature of the QPM to reject sperm cells which are non-compliant with the WHO2021 criteria.

ICSI success rates are highly dependent on clinical parameters, such as female age and oocyte quality as well as male age and sperm quality [28]. In conventional ICSI, a live, swimming and rotating sperm cell needs to be selected by the embryologist without access to a quantitative, objective evaluation of the cell’s morphology [12,29]. The proposed system, the QPM, was designed to address one of the causes for the relatively low per-cycle success rate of ICSI, by providing embryologists with a quantitative, objective analysis of the sperm cell prior to its injection into the oocyte [21]. The great variance between the a priori and a posteriori determination of normal sperm cell morphology, as demonstrated in this study, highlights the need for a better filter to assist embryologists in making evidence-based decisions when selecting sperm cells for use in ICSI.

Sperm morphology has been considered an indicator of male fertility and success with assisted reproductive technologies; men with fertility problems have a lower percentage of morphologically normal cells when compared to men with proven fertility. Moreover, early studies have shown that the morphology of spermatozoa bears a definite relation to the success of their ability to migrate through the cervical mucus as cells with enlarged and irregular heads are blocked by the selective hazard of cervical mucus. If done correctly and with strict application of existing guidelines as outlined in the WHO2021 guidelines, sperm morphology measurement can play an important role in the clinical evaluation of male fertility. However, this is a mechanism that is not fully understood. Spermatozoa found in the cervical mucus at the level of the internal os are usually an apparently homogeneous population, in contrast to the spermatozoa found in the seminal pool [30,31,32]. The measurements of head length, head width, head length–width ratio and acrosome-to-head ratio were adopted by the WHO strict criteria as an international gold standard. Sperm cell deviation from such dimensions is correlated with low natural pregnancy, IUI, IVF and ICSI outcomes and are correlated with a high degree of DNA fragmentation [33,34,35,36,37].

With the progressive description of sperm abnormalities, the WHO percentages for normal sperm decreased dramatically [38]. In the first edition of the WHO manual, the average normal morphology was 80.5% [39], which decreased to 50% in the second edition [40], 30% in the third edition [41] and 14% in the fourth edition [42], and is currently 4% for the fifth and sixth editions [24,43]. For the third edition of the WHO manual, the Tygerberg strict criteria were implemented. For these criteria, sperm with “borderline” abnormal features were classified as abnormal. It is commonly believed that the decline in reference values is mostly due to the introduction of strict criteria [43] but also by a decline in the number of morphologically normal sperm due to changes in lifestyle and different environmental factors [1]. The evaluation of sperm morphology was and still is regarded as subjective since it must be performed manually and by the human eye. Even today, most of the modern-day computer-assisted sperm morphology analysis (CAMA) systems still largely depend on human operator skills and suffer from the same technical problems as manual sperm morphology evaluations [32]. Today, in most laboratories, the assessment of sperm morphology is subjective, qualitative or toxic to the sperm and thus cannot be used prior to ICSI [32,38]. In a study assessing inter-observer variability in the results of sperm morphology and sperm antibody levels from 20 different laboratories, there was wide variation. Between labs, sperm morphology measurements have been shown to vary, and 40% of labs had a coefficient of variance/variation (CV) between 10 and 20%, and three labs had a CV >  20%, indicating wide inter-lab variability [44]. In our study, we present a solution that can use the consensus criteria for sperm morphology without fixing or staining the sperm cells. Our results showed that a QPM method used with the Q300™ system is fast, objective, quantitative and non-toxic and displays high repeatability. This method can be used before ICSI to better select sperm cells that comply with the strict WHO2021 criteria. Importantly, although ICSI could bypass physiologic sperm selection and abnormal sperm morphology, as it compensates for many steps of sperm fertilization, including swimming to the oocyte (motility), binding to the zona pellucida and the acrosome reaction, it includes artificial sperm selection by the embryologist as a crucial step that affects the clinical outcome [32,38] and thus should be under continuous improvement.

## 5. Conclusions

In conclusion, it was demonstrated that the Q300™ can provide accurate measurements of live sperm cells and inform embryologists in the selection of the most appropriate cells (based on the dimensional criteria of the WHO2021 guidelines) without chemical staining. Based on our previously published data, a next-generation classifier may be based on integration of clinical outcome data and characteristics such as sperm DNA-fragmentation, dry mass, 3D density, shape, motility, volume, and intracellular organization for self-improving evidence-based criteria to select sperm for ICSI.

The strengths of our study are that it is a multi-center randomized clinical trial (RCT) and a double-blinded prospective study. The data were from sperm samples collected from patients already at IVF clinics for infertility treatment and not from known fertile patients in order to address the unmet need for a better sperm selection method used during IVF treatment. A limitation of our study is that the data represent only 326 sperm cells. In future studies, we aim to image and analyze a larger set of sperm cells to be injected during IVF treatment. A second limitation is that this study examines only morphological analyses. While this approach has the potential to significantly improve the ICSI fertilization success rate, combining individual motion-analytics with the proposed morphological evaluation may lead to even greater compliance with WHO2021 guidelines and other consensus criteria for the selection of individual sperm cells for ICSI. Further studies are needed to evaluate the impact of using such a system on the embryo quality, percent of viable/usable embryos and in situ clinical outcome measures.

## Figures and Tables

**Figure 1 biomedicines-11-02614-f001:**
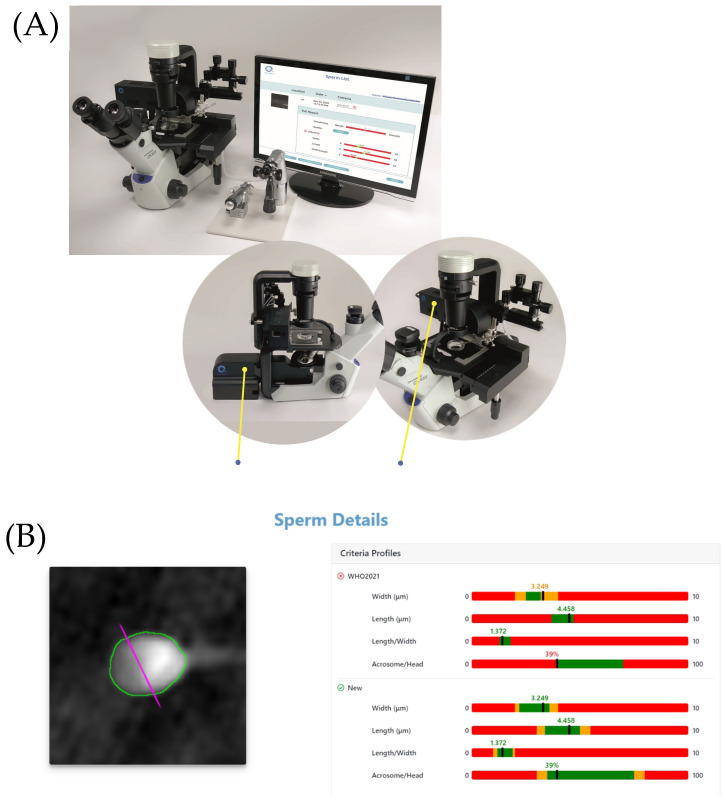
The Q300^TM^ system (**A**) and an image of the QPM results screen (**B**).

**Figure 2 biomedicines-11-02614-f002:**
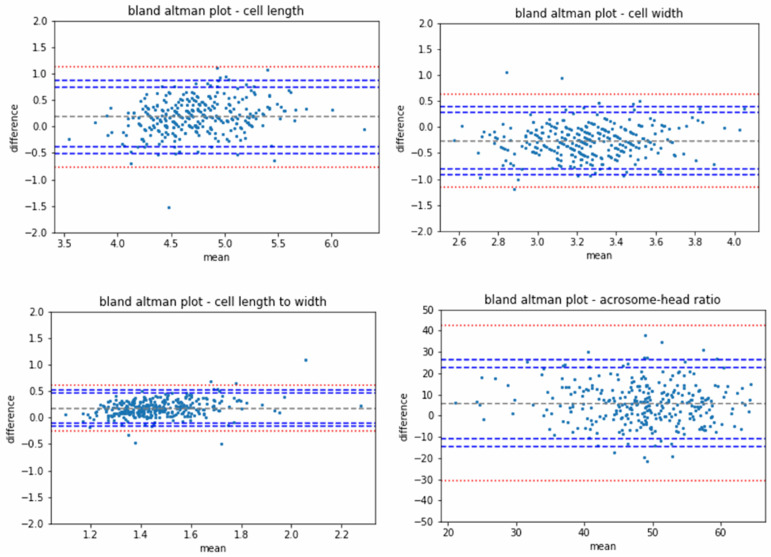
Agreement between QPM results and the reference method is demonstrated in Bland–Altman plots. Each dot in the graph represents the absolute difference between the two modalities for a specific sperm cell and for the specific measured parameter. Confidence intervals are presented in blue, and maximum allowed limits are presented in red.

**Figure 3 biomedicines-11-02614-f003:**
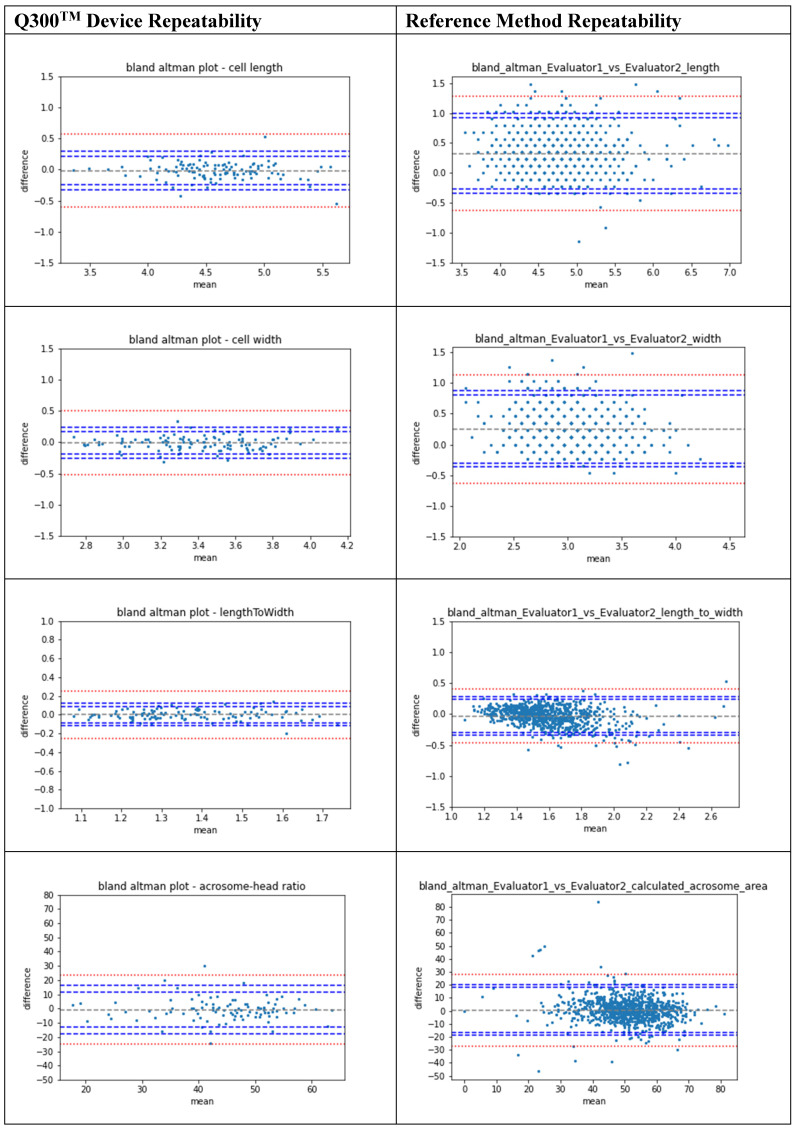
The agreement between QPM repeatability results and the reference method repeatability results demonstrated in Bland–Altman plots. Each dot in the graph represents the absolute difference between the two measurements of the same parameter measured for the same cell for evaluation of repeatability. Confidence intervals are presented in blue.

**Table 1 biomedicines-11-02614-t001:** Research sites and ethical approval.

Site Name	Ethic Committee and Address	Approval Date	Approval Number
Meir Medical Center	IRB-Helsinki Committee Meir Medical Center, Tschernihovski 59, Kfar Saba, Israel	28 July 2021	0121-21-MMC
Wolfson Medical Center	IRB-Helsinki Committee Wolfson Medical Center, 62 Halohamim Street, Holon, Israel	31 August 2021	0121-21-WOMC
Barzilai University Medical Center	IRB-Helsinki Committee Barzilai University Medical Center, 2 Hahistadrout Street, Ashkelon, Israel	23 March 2022	0113-21-BRZ

**Table 2 biomedicines-11-02614-t002:** Normal morphology ranges for the above endpoints recommended by WHO.

Sperm Parameter	WHO2021 Normal Range
Head length of a sperm cell (μm)	3.7–4.7
Head width of a sperm cell (μm)	2.5–3.2
Length-to-width ratio	1.3–1.8
Acrosome-area-to-head-area ratio × 100 (%)	40–70

**Table 3 biomedicines-11-02614-t003:** Reasons for sperm cell exclusion from evaluation for both modalities.

QPM—Reason for Excluding Procedures	
Cell with insufficient movement	38 (2.6%)
Inadequate focus	593 (41%)
A total of 631 sperm cells were excluded out of 1451 (43.6%)	
Reference (BF stained)—reasons for excluding images	
Inadequate focus	1699 (60.8%)
Unclear borders of cell nucleolus	126 (4.5%)
Cell was imaged in a position not appropriate for evaluation (not flat)	72 (2.5%)
A total of 1897 images were excluded out of 2791 images (68%), reflecting additional 820 sperm cells that were excluded beyond the 631 sperm cells that the device had excluded.	

**Table 4 biomedicines-11-02614-t004:** Baseline clinical characteristics and subject demographics.

Parameter	Statistic	Parameter	Statistic
Age	38.5 (STD-7.4)	History of Drug Abuse	21.9% (16/73)
Race	% (73/73)	Current	10.9% (8/73)
White	89% (65/73)	Former	10.9% (8/73)
Unknown	10.9% (8/73)	Medical Condition	% (73/73)
Ethnicity	% (73/73)	Current	32.8% (24/73)
Hispanic or Latino	1.7% (1/73)	None	67.1% (49/73)
Not Hispanic or Latino	76.7% (56/73)	Concomitant Medication	% (73/73)
Not Reported	4.1% (3/73)	Current	28.7% (21/73)
Unknown	17.8% (13/73)	None	71.2% (52/73)
Smoking	% (73/73)	Collected sperm sample	% (73/73)
Current	32.8% (24/73)	Yes	100% (73/73)
Former	20.8% (10/48)	No	0% (0/73)
None	52% (38/73)	Live Cell Evaluation and Analysis	% (73/73)
Drinks Alcohol	% (73/73)	Yes	95.8% (70/73)
Current	46.5% (34/73)	No	2.7% (2/73)
None	52% (38/73)		

## Data Availability

Not applicable.

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
