# Peer review of "Stain-Free Sperm Analysis and Selection for Intracytoplasmic Sperm Injection Complying with WHO Strict Normal Criteria"

_biomedicines, 2023, doi:10.3390/biomedicines11102614_

Round 1

Reviewer 1 Report

The authors carried out a stain-free sperm analysis and selection for ICSI complying with WHO strict normal criteria.

The manuscript is well-structured and discusses an important problem.

A major concern in this manuscript is the lack of use or evaluation of medicines in the study. As such, I am not convinced that it fits with the scopes of the journal. The authors, in collaboration with the editorial office, should consider transferring the manuscript to another relevant journal.

I remind everyone that the IJERPH lost their impact factor because of publishing articles outside the scope of that journal. DO the authors wish the same to happen to Biomedicines? Clarivate is very strict in maintaining standards and they are continuously on the lookout for breaches of terms of publication.

Apart from the above, I support publication. However, English language corrections must be performed throughout the manuscript.

If the issue of the suitability of the journal is resolved, then the manuscript can be accepted.

English language corrections must be performed throughout the manuscript.

Author Response

Dear Reviewer,

Thank you for your prompt response and valuable feedback on our manuscript titled "Stain-free sperm analysis and selection for ICSI complying with WHO strict normal criteria". We appreciate your time and effort in reviewing our work. We thank you for your valuable comments and suggestions. Changes in the revised manuscript are highlighted. Thorough revision was conducted as suggested. Changes were made upon comments as below (in bold):

In response to your comments:

  1. You marked that the introduction section "can be improved" and "provide sufficient background and including all relevant references".

Answer: The introduction section was thoroughly reorganized and extended and relevant references were added and included subjects such as infertility as a global health concern and the advantages and possibilities of full 3D refractive index imaging.

  1. You marked that the conclusion section "can be improved" and "support the results".

Answer: The conclusion section was revised and extended accordingly and thoroughly.

  1. You marked "extensive editing of English language required"

Answer: The manuscript was thoroughly reviewed by a native English speaker.

  1. A major concern in this manuscript is the lack of use or evaluation of medicines in the study.

QPM allows measuring the cell refractive-index distribution, thus using an intrinsic contrast mechanism of the cells to allow quick and reliable measurements of its internal morphology without using chemical staining, making it possible to apply the WHO-2021 criteria of normal morphology for ICSI treatment. We developed a useful tool for embryologists to select the best sperm cells during fertility treatments and hope the prestige journal of Biomedicines will acknowledge the scientific biomedical value of our results.

If you require any further information or clarification regarding these revisions, please do not hesitate to let us know. We are committed to ensuring that our manuscript meets the journal's standards.

Thank you again for your guidance, and we look forward to hearing from you soon.

Reviewer 2 Report

The authors are kindly requested to consider the following recommendations for their submitted manuscript:

- define all abbreviations where they are first appearing;

- write the equations by using the equation editor;

- include standard deviation values for the reported values;

- extend the discussion of the results (refer to the included graphs and compare them to other reported values on this subject);

- highlight the main advantages of the method.

Moderate English editing is necessary for the improvement of the manuscript.

Author Response

Dear Reviewer,

Thank you for your prompt response and valuable feedback on our manuscript titled "Stain-free sperm analysis and selection for ICSI complying with WHO strict normal criteria". We appreciate your time and effort in reviewing our work. We thank you for your valuable comments and suggestions. Changes in the revised manuscript are highlighted. Thorough revision was conducted as suggested. Changes were made upon comments as below (in bold):

In response to your comments:

  1. Define all abbreviations where they are first appearing.

Answer: The manuscript has been carefully reviewed and the missing information was added. E.g. in the introduction section: "Intra cytoplasmic sperm injection (ICSI) is a specialized assisted reproductive technology (ART) procedure used to assist in cases of infertility when male factors are the primary cause [2]".

  1. Write the equations by using the equation editor.

Answer: The equation editor has been used in the Results section:

For example, in the length plot, the maximum allowed limits located at:

  1. Include standard deviation values for the reported values.

Answer:

  1. Standard deviation values were added in the abstract: "Lines 25-28: Comparison of the two modalities resulted in mean difference of 0.18 µm (CI -0.442-0.808 µm, 95%, STD – 0.32 µm) for head length, -0.26 µm (CI -0.86-0.33µm, 95%, STD – 0.29 µm) for head width, 0.17 (CI -0.12-0.478, 95%, STD – 0.15) for Length-width ratio and 5.7 for acrosome-head areas ratio (CI -12.81-24.33, 95%, STD – 9.6)"
  2. Standard deviation values were added in the Results section: "Agreement of the two modalities resulted in bias of 0.18 µm (CI -0.442-0.808 µm, 95%, STD – 0.32 µm) for head length, -0.26 µm (CI -0.86-0.33µm, 95%, STD – 0.29 µm) for head width, 0.17 (CI -0.12-0.478, 95%, STD – 0.15) for length-width ratio and 5.7 for acrosome-head areas ratio (CI -12.81-24.33, 95%, STD – 9.6). Bland Altman plots are presented in Figure 2" and "Repeatability Agreement of the QPM vs. the reference method for each parameter resulted in a bias of -0.017 µm (CI -0.28-0.12 µm, 95%, STD – 0.13 µm) vs. 0.32 µm (CI -0.3-0.95 µm, 95%, STD – 0.32 µm) for head length respectively, -0.012 µm (CI -0.22-0.09 µm, 95%, STD – 0.1 µm) vs. 0.25 µm (CI -0.33-0.83 µm, 95%, STD – 0.29µm) for head width, 0.0006 (CI -0.102-0.05, 95%, STD - 0.05) vs. -0.03 (CI -0.3-0.25 95%, STD – 0.14) for Length-width ratio and bias of -0.73 (CI -15.1-6.7, 95%, STD – 7.49) vs. 0.65 (CI -17.5-18.8, 95%, STD – 9.29) for acrosome-head areas ratio. Bland Altman plots are presented in Figure 3".
  3. Standard deviation values were added in Table 4:

Parameter

Statistic

Age

38.5 (STD-7.4)

  1. Extend the discussion of the results (refer to the included graphs and compare them to other reported values on this subject).

Answer: The discussion section was revised and extended accordingly and thoroughly: "This study demonstrated that QPM technology can be successfully used in an IVF laboratory environment. Agreement between the QPM and the reference method was ad-equate and as can be seen in the Bland-Altman plots, the confidence intervals were well within the maximum allowed limits for all evaluated parameters. Agreement between the QPM and the reference method was performed for sperm head length, head width, head length-width ratio and acrosome to head ratio. A sensitivity analysis, performed between the results derived from the QPM and the reference method, aiming to reflect the ability of the QPM to detect the abnormal (non-complaint with WHO2021) sperm cells resulted in 88.6%, the accuracy of the method was 73.4%. A sub analysis of sensitivity and accuracy performed only on the population which were a priori evaluated as normal by the manual selection resulted in 85.2% sensitivity and 71.5% accuracy. Moreover, Repeatability of measurements was significantly higher in the QPM modality compared to the reference.

Another potentially important finding from the study was that less than 25% of the cells selected by embryologists through a manual and subjective selection were later con-firmed to be compliant with WHO2021 criteria. This was evident both when comparing the manually selected sperm cells via results derived by QPM and by the reference method results. As shown by the sensitivity analysis performed between the two modalities for detecting abnormal sperm cells, the QPM results were satisfying with a high degree of sensitivity and accuracy. This is arguably the most useful feature of the QPM, to reject sperm cells which are non-compliant with WHO2021 criteria.

ICSI success rates are highly dependent on clinical parameters, such as female age and oocyte quality as well as male age and sperm quality [28]. In conventional ICSI, a live, swimming, and rotating sperm cell needs to be selected by the embryologist, without access to a quantitative, objective evaluation of the cell’s morphology [29,12]. The proposed system, the QPM, was designed to address one of the causes for the relatively low per-cycle success rate of ICSI, by providing embryologists a quantitative, objective analysis of the sperm cell, prior to its injection into the oocyte [21]. The great variance between the a priori and posteriori determination of normal sperm cell morphology, as demonstrated in this study, highlights the need for a better filter to assist the embryologist ins making an evidence-based decision when selecting sperm cells for use in ICSI.

Sperm morphology has been considered an indicator of male fertility and success with assisted reproductive technologies subfertile men have a lower percentage of morphologically normal cells when compared to men with proven fertility. Moreover, early studies have shown that morphology of spermatozoa bears a definite relation to the success of their ability to migrate through the cervical mucus as cells with enlarged and irregular heads are blocked by the selective hazard of cervical mucus. If done correctly and with strict application of existing guidelines as outlined in the WHO2021 guidelines, sperm morphology measurement can play an important role in the clinical evaluation of male fertility. However, this is a mechanism that is not fully understood. Spermatozoa found in the cervical mucus at the level of the internal os are usually an apparently homogeneous population, in contrast to the spermatozoa found in the seminal pool [30-32]. The measurements of head length, head width, head length-width ratio and acrosome to head ratio were adopted by the WHO strict criteria as an international gold-standard. Sperm cell deviation from such dimensions is correlated with low natural pregnancy, IUI, IVF, and ICSI outcomes and are correlated with high degree of DNA fragmentation [33-37].

With the progressive description of sperm abnormalities, the WHO percentages for normal sperm decreased dramatically [38]. In WHO 1st edition, the average normal morphology was 80.5% [39], which decreased to 50% in the 2nd edition [40], 30% in the 3rd edition [41], 14% in the 4th edition [42], and is currently 4% for the 5th and 6th edition [24,43]. For the 3rd edition of the WHO manual, the Tygerberg strict criteria were implemented. For these criteria, sperm with “borderline” abnormal features were classified as abnormal. It is commonly believed that the decline in reference values is mostly due to the introduction of strict criteria [43], but also by a decline in the number of morphologically normal sperm due to changes in lifestyle and different environmental factors [1]. The evaluation of sperm morphology was and still is regarded as subjective since it must be done manually and by the human eye. Even today most of the modern-day Computer-Assisted sperm Morphology Analysis (CAMA) systems still largely depend on human operator skills and suffer from the same technical problems as manual sperm morphology evaluation [32]. Today in most laboratories, the assessment of sperm morphology is subjective, qualitative, or toxic to the sperm and thus cannot be used prior to ICSI [32,38]. In a study assessing inter-observer variability of results of sperm morphology and sperm antibody levels from 20 different laboratories, there was wide variation. Between labs, sperm morphology measurements have been shown to vary, and 40% of labs had a coefficient of variance/variation (CV) between 10 and 20%, and 3 labs had a CV > 20%, indicating wide inter-lab variability [44]. In our study we present a solution that can use the consensus criteria for sperm morphology without fixing or staining the sperm cells. Our results showed that a QPM method used by the Q300™ system is fast, objective, quantitative, non-toxic, and displays high repeatability. This method can be used before ICSI to better select sperm cells that comply with the strict WHO2021 criteria. Importantly, although ICSI could bypass physiologic sperm selection and abnormal sperm morphology, as it compensates for many steps of sperm fertilization, including swimming to the oocyte (motility), binding to the zona pellucida, and the acrosome reaction, it includes artificial sperm selection by the embryologist as a crucial step that affects the clinical outcome [32,38] and thus should be under continuous improvement".

  1. Highlight the main advantages of the method.

Answer: The strength of the study was added to the conclusion section: " The strengths of our study are that it is multi-center randomized clinical trial (RCT) and a double blinded prospective study. The data was from sperm samples collected from patients already at IVF clinics for infertility treatment, and not from known fertile patients in order to address the unmet need for better sperm selection method used during IVF treatment".

  1. You marked - moderate English editing is necessary for the improvement of the manuscript.

Answer: The manuscript was thoroughly reviewed by a native English speaker

  1. You marked - improving the Introduction, providing sufficient background and including all relevant references.

Answer: The introduction section was reorganized and extended. The reference to infertility as a global health concern was added. In addition, the advantages and possibilities of full 3D refractive index imaging were shown.

  1. You marked - citing references relevant to the research.

Answer: New citations were added to the manuscript. E.g. Unfortunately, the data concerning the utilization of HA in ICSI is still controversial [15].

  1. You marked - improving the research design and methods.

Answer: The methods and the design sections were extended accordingly and the following text was added: "This study is a double blinded comparison between the automated measurements of the QPM technology used in the QPM system and the corresponding measurements of the same sperm cells after they were chemically stained in accordance with WHO human sperm processing protocols [25] and acquired via standard bright field microscopy (BFM) imaging [27]. Imaging and analysis were performed by senior embryologists as the hu-man operators. The objective of this comparison is to explore the agreement between the QPM and a reference method.

Since the study compares morphological measurements of live, motile sperm cells, these parameters can deviate somewhat when derived from different intervals of the same QPM recording. Therefore, an additional objective of this study was to assess the repeatability of morphological measurements of the same sperm cells, when taken from two different intervals of the same recording. A similar repeatability assessment was performed between two sets of BFM images of sperm cells, and manually measured by the readers. A comparison between the repeatability of the QPM and the reference method was also per-formed.

Since the QPM system is designed to aid the embryologist in confirming compliance of specific sperm cells with consensus morphometric criteria, it is imperative to verify that the QPM system can reliably measure such parameters even with non-WHO2021-compliant sperm cells. For this reason, the sample group was a priori di-vided into an equal number of “normal morphology” and “abnormal morphology” sperm cells, as subjectively decided by the embryologists.

Another study endpoint was to assess the accuracy of the subjective classification of sperm cells, as performed by embryologists, in reference to WHO2021 morphometric criteria. Put simply, how likely is a sperm cell that an embryologist identifies as displaying “normal morphology” to be verified as having such morphology through QPM virtual staining and chemical staining".

  1. You marked - improving the Results described.

Answer: We added the standard deviation values in order to make sense of data and draw meaningful conclusions from the research findings.

  1. You marked - improving the discussion section.

Answer: The discussion section was revised and extended accordingly and thoroughly. Please see our answer in point 4.

If you require any further information or clarification regarding these revisions, please do not hesitate to let us know. We are committed to ensuring that our manuscript meets the journal's standards.

Thank you again for your guidance, and we look forward to hearing from you soon.

Round 2

Reviewer 2 Report

The authors have correspondingly addressed all the recommendations of the reviewer.